# Initial Bacterial Adhesion and Biofilm Formation on Aligner Materials

**DOI:** 10.3390/antibiotics9120908

**Published:** 2020-12-15

**Authors:** Sibel Tektas, Thomas Thurnheer, Theodore Eliades, Thomas Attin, Lamprini Karygianni

**Affiliations:** 1Clinic of Orthodontics and Paediatric Dentistry, Center of Dental Medicine, University of Zurich, 8032 Zurich, Switzerland; sibel.tektas@zzm.uzh.ch (S.T.); theodore.eliades@zzm.uzh.ch (T.E.); 2Clinic of Conservative and Preventive Dentistry, Center of Dental Medicine, University of Zurich, 8032 Zurich, Switzerland; thomas.thurnheer@zzm.uzh.ch (T.T.); thomas.attin@zzm.uzh.ch (T.A.)

**Keywords:** orthodontic appliances, aligners, initial microbial adhesion, oral biofilm

## Abstract

The present study aims to assess the initial bacterial adhesion and biofilm formation on different aligner materials. A total of four different aligner materials, CA-medium (CAM), copolyester (COP), Duran (DUR), Erkodur (ERK), were tested. Stimulated human saliva was obtained from six healthy volunteers. Salivary bacteria were harvested by centrifugation, and 1 mL of the salivary suspension was injected onto each sample surface for 2 h and 3 days, respectively. The samples were then washed twice with 5 mL 0.9% NaCl solution, and non-adherent bacteria were removed. The adherent microorganisms were dislodged from the sample surfaces after ultrasonication for 4 min in 1 mL 0.9% NaCl on ice. After the incubation of the adherent salivary bacteria under both aerobic and anaerobic conditions on Columbia blood agar plates at 37 °C and 5% CO_2_ and in anaerobic jars overnight, several dilutions thereof were used for the determination of CFUs. This protocol was applied three times, obtaining an average of nine independent measurements for each material group. Overall, the differences between the tested aligner materials as well as between the materials and controls were not of statistical significance (*p* > 0.05). Regarding initial bacterial attachment and biofilm formation, the tested aligner materials are comparable to enamel and metal orthodontic brackets and can be therefore considered for clinical use. The four tested aligner materials CAM, COP, DUR, ERK showed no significant differences in initial microbial attachment and biofilm formation of aerobic and anaerobic species compared to enamel and conventional brackets.

## 1. Introduction

In recent years, the public demand for esthetic and metal-free orthodontic devices as an alternative to treatment with fixed appliances has been increased [1]. Due to their invisibility and, therefore, better appearance, aligners have gained in popularity [2].

At the end of 1990s the first aligner, namely Invisalign, a polyurethane-based material, was introduced by Align Technology (San Jose, CA, USA) [3]. With the aid of a software program, a series of transparent removable aligners are designed to move the teeth step-by-step towards a predefined position. It is highly advised that the aligners are worn 20–22 h per day, and that should be removed only during food intake and tooth brushing [4]. Aligners are not considered to interfere with oral hygiene [5] since patients must clean surfaces of reduced complexity and with very few retentive sites for bacterial colonialization [6].

The presence of microbial biofilms is etiologically associated with the onset of oral diseases such as caries and periodontitis. Dental plaque, also known as oral biofilm, has been described as a complex network of microbial cells embedded in extracellular polymers of microbial origin that mainly attaches to tooth surfaces or gingival epithelial cells [7]. Bacteria do not adhere directly to tooth structures [8] but indirectly through their adhesion to the pellicle, an acellular proteinaceous layer covering enamel surfaces [9]. The oral microflora, which can be resistant to different antimicrobial agents, consists of a wide range of different microbial species (up to 700), including, among others, aerobic and anaerobic bacteria [10].

Not only tooth surfaces but also foreign surfaces, such as orthodontic devices, can represent an ideal substrate for biofilm formation [11]. To date, it has been reported that recessed and sheltered areas of aligner splints, such as cusp tips and attachment dimples, show higher susceptibility to biofilm formation than flat surfaces [12]. Surface morphology, surface chemistry and surface charge are mainly responsible for bio-adhesion and plaque formation, which could initiate a cascade of events leading to dental hard tissue or periodontal soft tissue disease. Hence, not only the design of the aligner but also the material structure, which can affect bacterial adhesion, also has an impact on the oral microenvironment [13,14].

Very little data are available in the literature concerning biofilm formation on aligner materials. Since biofilms are considered as potential indicators for oral infections, a deeper understanding of their initiation, formation and spread on the aligner materials is of major clinical importance. Thus, this study aims to examine the initial microbial attachment and biofilm formation on various thermoplastic aligner materials with the same chemistry but different surface characteristics. The null hypothesis of the study is that there are no significant differences in initial microbial adhesion and biofilm formation between the tested aligner samples and controls.

## 2. Results

### 2.1. The Initial Adhesion of Aerobic and Anaerobic Microorganisms on Aligners Was Comparable to That on Enamel and Brackets

Figure 1A shows the results of the initially adherent oral aerobic microorganisms on the four tested aligner materials (COP, CAM, ERK, DUR) after a 2 h incubation in human saliva, plus the control samples (enamel, metal brackets). No significant differences could be found in the viable bacterial count after 2 h of initial microbial adhesion between the tested aligners and the control groups.

Regarding the initially adherent oral aerobic microorganisms, in particular, enamel, which served as a control, revealed a log_10_ CFU value of 3.3 ± 1.8 log_10_ (median 3.3), while the CFU counts in the presence of brackets were 2.9 ± 1.6 log_10_. The CFU counts of COP were 3.3 ± 1.8 (median 3.3) on a log_10_ scale, while CAM showed a mean of 2.7 ± 1.5 log_10_ (median 2.9). The analysis of ERK (2.4 ± 2.6 log_10_ CFU, median 2.1) and DUR (3.8 ± 1.7 log_10_ CFU, median 4.1) yielded similar results.

Figure 1B shows the numbers of viable anaerobic microorganisms on the thermoplastic surfaces of the tested aligner materials. The CFU counts of COP (3.8 ± 1.3 log_10_, median 3.5), CAM (2.6 ± 1.8 log_10_, median 2.6), ERK (2.1 ± 1.9 log_10_, median 2.4) and (DUR 4.0 ± 1.7 log_10_, median 4.4) yielded a comparable outcome after 2 h of incubation in human saliva. No significant differences could be found when compared to tested control groups enamel (4.4 ± 2.0 log_10_ CFU, median 5.1) and brackets (2.9 ± 1.7 log_10_ CFU, median 2.3).

### 2.2. Aligner Materials and Controls Presented Similar Biofilm Formation

The CFU values of aerobic and anaerobic bacteria on each aligner material and controls after 72 h of biofilm formation are depicted in Figure 2. Despite the increased amounts of CFUs, similar results could be found as for the initial adhesion after a 2 h incubation.

The CFU counts of aerobic microbes on the investigated aligner materials were the following: COP (6.9 ± 1.6 log_10_, median 7.4), CAM (6.7 ± 1.2 log_10_, median 6.4), ERK (6.1 ± 1.7 log_10_, median 6.9) and DUR (6.6 ± 1.3 log_10_, median 7.2). Compared with the control samples, enamel and metal brackets exhibiting CFU values of 6.8 ± 0.9 log_10_ (median 6.9) and 5.8 ± 1.2 log_10_ (median 6.0), respectively, no statistical differences could be shown (Figure 2A).

Regarding the biofilm formation of oral anaerobic microorganisms on control samples, a log_10_ CFU value of 6.9 ± 1.0 log_10_ (median 7.1) was detected on enamel, while brackets showed 6.1 ± 1.0 log_10_ (median 6.1) of CFU counts. The comparison of CFU counts of the control samples with COP (7.1 ± 1.7 log_10_, median 7.6), CAM (6.9 ± 1.2 log_10_, median 6.7), ERK (6.5 ± 2.0 log_10_, median 7.4) and DUR (7.3 ± 1.4 log_10_, median 7.7) was not of statistical significance (Figure 2B).

## 3. Discussion

Regarding the surface properties in the context of initial microbial adhesion, the four aligner materials showed similar results when compared with both orthodontic metal brackets and enamel. Enamel served as a control for the comparison of the natural tooth surface with the PETG-based surfaces. Brackets represented the conventional orthodontic surfaces and are therefore suitable controls for comparison with modern orthodontic materials, namely the PETG-based surfaces. Interestingly, even after three days of incubation and biofilm formation, no differences could be found between the tested groups and the controls. Independent of the time (2 h, 72 h), the comparison of aerobic and anaerobic bacteria revealed no difference in the amount and distribution of the microbial population between the tested material- and control surfaces. Based on the experimental results of this study, the null hypothesis was confirmed since there are no significant discrepancies in initial microbial adhesion and biofilm formation between the tested aligner samples and controls. To our knowledge, this is the first study to assess the initial bacterial adhesion and biofilm formation on the four different aligner materials, which are commercially available.

It is well-known that oral bacteria attach to a surface indirectly through the pellicle [15]. The salivary pellicle is a protein-containing, bacteria-free film, which covers tooth surfaces [16] and mainly serves as a site for bacterial attachment, which is regarded as a prerequisite for the bacterial colonization of the tooth- and other oral surfaces [17]. The related literature demonstrates interesting findings in regard to bacterial colonization in the oral cavity and underlines the complexity of microbial attachment on different surfaces. The microbial colonization in the oral cavity includes the transport of microbial cells to the target surface, the initial microbial adhesion involving both a reversible and an irreversible stage, and further microbial attachment mediated by specific interactions [18]. The adhesion of microorganisms to soft tissue, hard tissue and material surfaces is the result of a wide array of parameters, including specific lectin-like reactions [17], electrostatic interactions and Van der Waals forces [18], which lead to the formation of biofilm [17]. Different surfaces (teeth, fixed appliances, dental implants, or removable appliances) are available for microbial colonization in the oral cavity, all with different surface characteristics [17].

After 72 h of biofilm formation on the tested aligner samples and controls, the amount of CFUs increased compared to microbial incubation of the material surfaces and the control for 2 h. Interestingly, at each time point (2 h, 72 h), no significant differences could be detected between the tested groups and the controls. Plaque formation, also known as biofilm formation, is a dynamic process involving several overlapping phases [7]. Dental plaque or oral biofilm adheres to teeth and consists primarily of microbes embedded in a polymeric matrix of proteins, peptides and polysaccharides [19]. Within dental plaque, microorganisms act as a community [20], building a complex biofilm upon accumulation on different oral surfaces, like enamel and artificial substrates [21] such as aligners and brackets [11,22]. Quirynen and Bollen have documented the acceleration of plaque formation with an increase mainly in the amplitude of surface roughness. The surface roughness of such a material is crucial in the development of an irreversible biofilm since it may create niches in which bacteria are protected from oral hygiene and shear forces.

Surface properties such as wettability, which are directly linked to bacterial adhesion, are influenced by the related physical surface characteristics [23,24]. The generated amplitude-, hybrid-, and spatial data comprised five individual parameters, namely average surface roughness (Sa), ten-point average roughness (Sz), root mean square surface roughness (Sq), developed surface area ratio (Sdr), and maximum pit height (Sv). Further definitions for each surface characteristic can be found in the study by Crawford et al. [25]. In literature, several studies report inconsistent outcomes in regard to the impact of Sq on bacterial adhesion [26]. Due to the fact that Sa can reportedly affect microbial attachment in the presence of Sq values higher than 0.2 μm, no significant differences could be observed for the tested aligner materials with significantly lower Sa values (8.1–9.6 nm). Since it is well known that smooth surfaces with a surface roughness (Sa) threshold of 0.2 μm inhibit bacterial adhesion, we decided to test PETG-based surfaces with similar Sa values (8.1–9.6 nm) in order to investigate if differences in other surface characteristics, e.g., Sz could affect microbial adhesion and biofilm formation on the smooth PETG-based surfaces. This assumption was not confirmed in our study.

Most manufacturers produce aligners with polyethylene terephthalate glycol (PETG), which is a non-crystallizing transparent amorphous copolymer with no strain-induced crystallization materials [27]. Only PETG-based samples were used in the present study, and that is probably the reason why they showed similar results in regard to CFU counts. Previous research showed all the PETG aligners tested were partially wetted by human saliva, with CAM demonstrating the least wettability [23]. Despite the fact that various variations of PETF aligner materials demonstrated similar surface morphology, surface roughness was found to strongly interfere with wetting phenomena. The tested materials showed differences in the total work of adhesion (WA), which represents the retention capacity of the intraoral film developed onto these materials, with higher work being associate with higher retention. Therefore, it may be postulated that differences in WA have a clinical impact within the context of pellicle formation [23]. The changes accompanying the intraoral service of aligners have been reported for a wide array of materials, including PETG and polyurethane-based polymers, with respect to mechanical properties and surface roughness in recent research from our group [28]. Because of the relatively short exposure time of aligners to the intraoral environment, the imposed changes are limited to the effects of water on the polymeric material. Hydrolytic degradation and attack of the polyurethanes by water have been described in the biomedical literature as the main mechanism of the disintegration of polymers exposed to aqueous media [29].

Since the main goal of the study was to examine the initial microbial adhesion and biofilm formation on different PETG-based aligner materials, the determination of CFUs was the best standard method to quantify the total cell count of the viable adherent microorganisms without focusing on specific bacterial species. The study design allowed a separate measurement of aerobic and anaerobic species since they have different pathogenic potential in general. Specifically, the supragingival plaque is dominated by aerobic species as the adherent bacteria are directly exposed to the open atmosphere of the oral cavity. This leads to an increase of cariogenic species [10]. It is well known that dental plaque becomes increasingly anaerobic in the presence of an anaerobic environment [30]. The low (even zero) CFU values during initial adhesion can be attributed to the presence of mainly non- or loosely adherent microorganisms on the tested PETG-based surfaces, which are extremely smooth with low Sa values (8.1–9.6 nm) inhibiting initial microbial adhesion. The persistence of the oral biofilm on the tooth causes inflammation of the surrounding gingival tissues, eventually facilitating the process of periodontal pocket formation [10]. The subgingival plaque contains mostly anaerobic organisms and is responsible for the initiation and establishment of periodontal disease [31].

Long-time clinical observation of orthodontic patients confirms these findings. Orthodontic appliances, particularly fixed appliances, lead to a change of the microenvironment for oral microbes. The presence of an acidic pH within dental plaque can be measured and triggers the growth of pathogenic species and, thus, a shift to a pathogenic oral microflora [32,33]. Furthermore, the risk of developing white spot lesions because of demineralization in enamel is significantly higher during orthodontic treatment. Additionally, if the patient shows poor oral hygiene, a higher incidence of white spot lesions can be found [34]. Similar observations have been made regarding caries development in dentin [35,36]. The inflammation of oral soft tissues is also associated with plaque formation [20,37] around brackets, which in turn leads to a commensal-to-pathogen transition of the oral microflora [38].

In regard to oral health, aligners are considered to have advantages in comparison to fixed appliances regarding oral health. Abbate et al. [4] reported that teenagers treated with aligners display less plaque and gingivitis. This agrees with previous data indicating that aligner therapy entails significantly lower plaque and gingivitis scores [39]. This may be attributed to the fact that aligners are removable, allowing the practice of good oral hygiene [5]. Regarding periodontal health as well as quantity and quality of adherent dental plaque, aligners were shown to be superior compared with traditional bracket systems in an earlier report [40]. Periodontal health indexes were significantly decreased during aligner therapy. Moreover, if one considers basic periodontal parameters during orthodontic therapy, aligners demonstrated significantly lower levels of plaque indices in contrast to lingual fixed appliances [5]. However, a current clinical trial detected no significant distinctions in the oral hygiene levels among patients treated with aligners or fixed appliances after an 18 month period of treatment [41]. This finding agrees with earlier clinical studies reporting no increased risk for periodontal disease after application of fixed appliances as well as removable aligners [42].

The in vitro character of this investigation may not reliably represent the complex environmental milieu present in the oral cavity. Thus, the long-term attachment of microbes to metal bracket surfaces cannot be reliably simulated by the 72 h immersion protocol. On the other hand, the use of human saliva as a medium facilitates the study of the actual attachment pattern and the comparative analysis among materials provides a useful insight for the anticipated attachment of microbes to different orthodontic materials. Further investigation of the biofilm characteristics on the PETG-based surfaces involves culture-based and molecular analyses of the microbial communities on species level as well as the biofilm metabolic activity.

## 4. Materials and Methods

### 4.1. Study Design

Pooled saliva collected from 6 donors were used to cultivate adherent salivary bacteria under both aerobic and anaerobic conditions. Four different aligner materials were tested: CA-medium (CAM), copolyester (COP), Duran (DUR), Erkodur (ERK). Four samples of each material group were incubated in saliva for 2 and 72 h, respectively. A bovine enamel sample and a bracket served as control groups. The number of cultivable bacteria was determined by counting the colony-forming units (CFUs).

### 4.2. Selection of Study Participants and Saliva Collection

Paraffin-stimulated saliva samples of six healthy volunteers (25 mL from each participant) of both sexes (30–35 years) were collected and pooled. Participants fulfilling one or more of the following criteria were not included in the study: smoking, infections or severe systemic conditions, e.g., salivary gland diseases, presence of carious lesions or periodontal disease, pregnancy or breastfeeding. The subjects refrained from eating or drinking for at least 1 h before saliva collection and from the intake of antibiotics, antimicrobial agents, e.g., chlorhexidine (CHX) 3 months prior to or during the study period. Immediately after its collection, the saliva was centrifuged for 5 min at 1000 rpm. The optical density of the saliva suspensions was assessed at 595 nm (Bio-Rad, Life Science Group, Hercules, CA, USA) against 0.9% saline solution (NaCl) as a blank. The supernatants were stored at −20 °C prior to use and served later as media and bacterial sources.

### 4.3. Test Specimen Preparation

#### 4.3.1. Aligner Samples

Four commercially available thermoplastic materials [23,27] for the production of orthodontic aligners were included in this study. The brand names, the manufacturers and the codes of all materials are presented in Table 1. Twelve thermoplastic sheets from each listed material were cut and removed from the labial surface of the central incisors. Round transparent specimens (diameter, 3 mm; width, 1 mm) were gained from the received disks with a metallic puncher made of hardened steel. Table 2 shows the surface different surface parameters of the tested aligner materials, which were described in our own recent study [23].

#### 4.3.2. Enamel Specimens and Metal Brackets

Enamel specimens were obtained from unscathed bovine incisors, which were stored in 0.1% thymol solution after extraction. Enamel cylinders (diameter, 4.5 mm; 27.2 mm^2^ surface area; height, 0.8 mm) were prepared out of the frontal surfaces of the bovine incisors of freshly slaughtered 2-year-old cattle using a water-cooled diamond core drill. Enamel specimens were embedded in acrylic resin blocks (6 mm in diameter; Paladur, Heraeus Kulzer, Hanau, Germany). The surfaces were ground flat with water-cooled silicon carbide paper disks (#1200, #2500, #4000, Gekko-Papier, Struers, Birmensdorf, Switzerland), followed by fine polishing with a 3- and 1-μm diamond abrasive under constant cooling (LaboPol-6, Struers, Birmensdorf, Switzerland). The disinfection protocol of the aligner and enamel involved the removal of the superficial smear layer by air drying and ultrasonication in 70% ethanol for 3 min. Thereafter, the disinfected samples were ultrasonicated twice in double-distilled water for 10 min and finally stored in distilled water for 24 h to hydrate before use. Metal orthodontic brackets (length, 4.2 mm; width, 3.1 mm; 40.76 mm^2^ surface area; height, 1.9 mm) were also used as controls. A total of 96 samples were tested in two different runs (8 samples for each material group at each time point).

### 4.4. Microbial Contamination of Aligner Materials and Controls

The thawed microbial suspension was diluted 1:2 with mFUM with 0.3% glucose. Twelve samples of each material surface (4 aligner materials, 1 enamel disk and 1 metal bracket as controls) were placed into multi-well plates (24-well plate; Greiner bio-one, Frickenhausen, Germany). Afterward, 1.6 mL of the microbial suspension was injected onto each sample surface. The 24-well plates were then incubated under anaerobic conditions at 37 °C for 2 h (initial microbial adhesion) and 72 h (biofilm formation), respectively. The medium (mFUM with 0.3% glucose) was changed after 16 h and 48 h to assess the biofilm formation after 72 h.

### 4.5. Quantification of the Adherent Oral Biofilm Microorganisms

After the incubation of the test surfaces and controls in microbial suspensions for 2 h and 72 h, respectively, the samples were washed thrice with 2 mL 0.9% saline solution (NaCl) to remove the non-adherent microorganisms and were subsequently transferred into a 5 mL tube with 1 mL NaCl [43]. The adherent microorganisms/biofilms were then dislodged from the aligner, enamel and bracket surfaces by sonification for 4 min (5 × 10% cycle, power 60%). Afterward, the suspensions of the aligners and controls were serially diluted up to 1:10^4^ in 0.9% NaCl with the aid of a spiral diluter (EDDY JET V.1.23, IG Instrumenten-Gesellschaft AG, Zurich, Switzerland). The final volume of the microbial suspension amounted to 1 mL. Subsequently, 50 μL of the dilutions were plated on Schedler plates (BD254042; Becton Dickinson, Heidelberg, Germany) so that aerobic and facultative anaerobic bacteria were cultivated at 37 °C and 5% CO_2_ for 5 days. Accordingly, anaerobic bacteria were also cultivated on Schedler plates at 37 °C for 10 days (anaerobic chamber; Anaerocult A; Merck, Darmstadt, Germany). The analysis was performed three times as multispecies experiments. The number of CFU per cm^2^ was counted by a single investigator. Each measurement was repeated twice. Two individual experiments were performed, and each group was represented in quadruplicate in each experiment.

### 4.6. Statistical Analysis

The Kolmogorov–Smirnov test was applied to investigate the normality of data distribution. Since the data were not normally distributed, a Kruskal–Wallis test was used to evaluate the differences between the controls and the experimental material groups. The significance level was set to *p* < 0.05. Values below the detection limit were ascribed the lowest detection limit value to allow for logarithmic transformation. The Prism v.8 statistical analysis software (GraphPad, La Jolla, CA, USA) was used to analyze the data.

## 5. Conclusions

It is of high clinical interest whether an orthodontic device causes plaque accumulation and, thus, oral diseases such as caries and gingivitis. In comparison to enamel surfaces and conventional metal brackets, the four tested aligner materials, CA-medium (CAM), copolyester (COP), Duran (DUR), Erkodur (ERK), showed no significant differences in initial microbial attachment and biofilm formation of aerobic and anaerobic species. These findings justify the assertion that these aligners are suitable for clinical use.

## Figures and Tables

**Figure 1 antibiotics-09-00908-f001:**
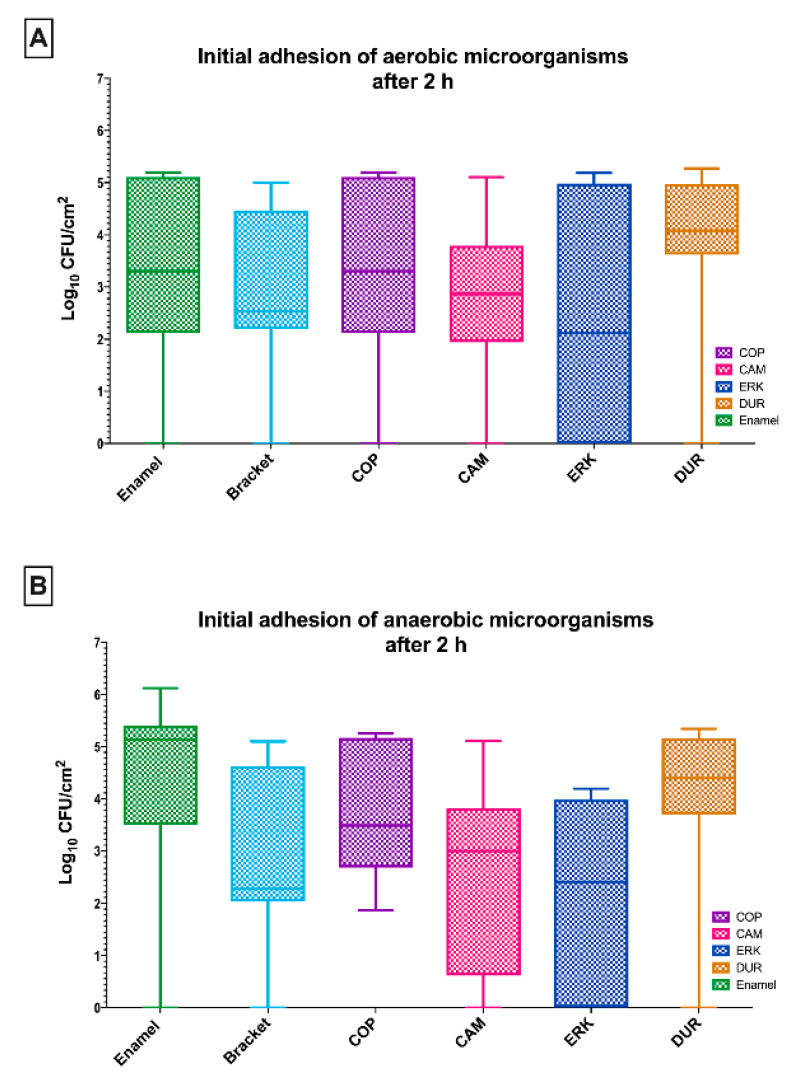
Boxplots depicting the colony-forming units (CFUs) of aerobic (**A**) and anaerobic (**B**) microorganisms on the tested aligner and control surfaces after an incubation time of 2 h. The medians, whiskers, and max/min outliers are demonstrated. Four different aligner materials were tested: CA-medium (CAM), copolyester (COP), Duran (DUR), Erkodur (ERK). Bovine enamel and metal brackets served as control groups.

**Figure 2 antibiotics-09-00908-f002:**
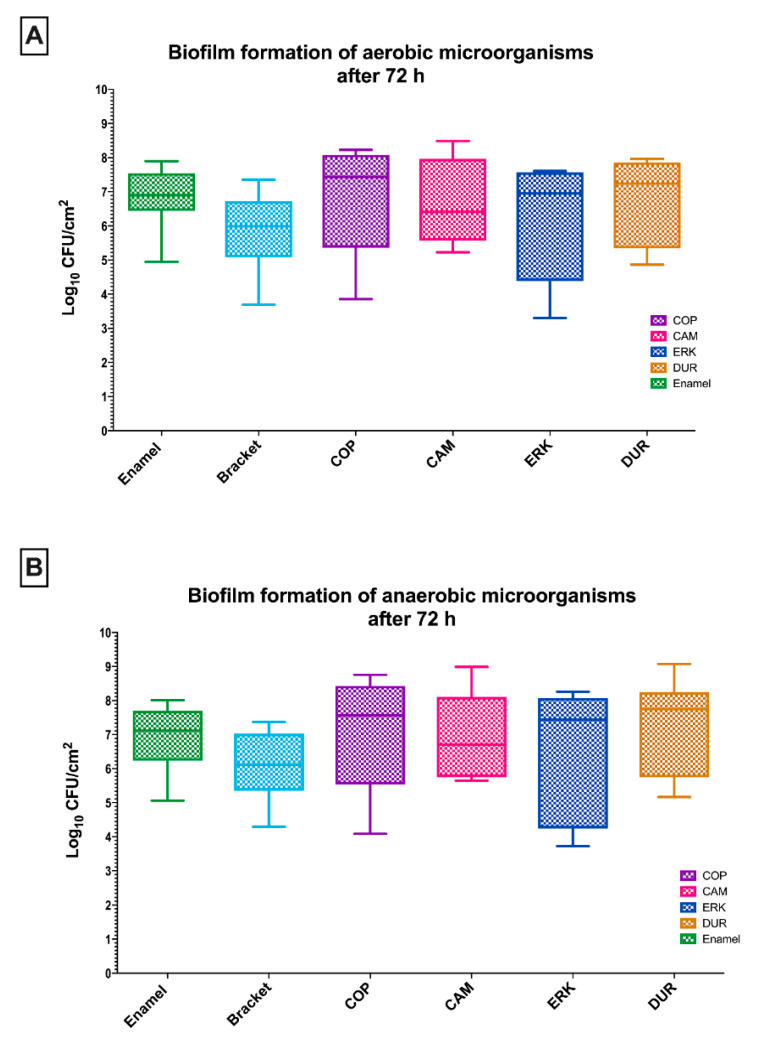
Boxplots depicting the colony-forming units (CFUs) of aerobic (**A**) and anaerobic (**B**) microorganisms on the tested aligner and control surfaces after an incubation time of 72 h. The medians, whiskers, and max/min outliers are demonstrated. Four different aligner materials were tested: CA-medium (CAM), copolyester (COP), Duran (DUR), Erkodur (ERK). Bovine enamel and metal brackets served as control groups.

**Table 1 antibiotics-09-00908-t001:** Brand names manufacturers, and codes for all materials tested.

Brand Name	Company	Code
CA-medium	Scheu-Dental, Iserlohn, Germany	CAM
Copolyester	Essix, Dentsply Raintree Essix Sarasota, FL, USA	COP
Duran	Great Lakes Dental Technologies, Tonawanda, NY, USA	DUR
Erkodur	Erkodent Erich Kopp, Pfalzgrafenweiler, Germany	ERK

**Table 2 antibiotics-09-00908-t002:** Mean values and standard deviations (in parentheses) of five surface roughness parameters, namely average surface roughness (Sa), ten-point average roughness (Sz), root mean square surface roughness (Sq), developed surface area ratio (Sdr), and maximum pit height (Sv).

	Surface Parameters				
Group	Sa (nm)	Sz (nm)	Sq (nm)	Sdr (%)	Sv (nm^3^/nm^2^)
CAM	8.1 (2.8) ^a^	54.8 (22.6) ^a^	13.1 (7.5) ^a^	0.002 (0.004) ^a^	1.0 (0.3) ^a^
COP	8.5 (0.6) ^a^	58.0 (3.5) ^a^	11.2 (1.2) ^a^	0.000 (0.000) ^a^	1.3 (0.2) ^a^
DUR	9.6 (2.5) ^a^	96.3 (24.1) ^a^	13.4 (4.8) ^a^	0.000 (0.000) ^a^	1.1 (0.2) ^a^
ERK	9.0 (2.0) ^a^	75.7 (26.4) ^a^	12.6 (3.2) ^a^	0.000 (0.000) ^a^	1.0 (0.2) ^a^

The superscript letter ^a^ denotes values without statistically significant differences per parameter within and between different treatment groups (*p* > 0.05).

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
