# Peer review of "Initial Bacterial Adhesion and Biofilm Formation on Aligner Materials"

_antibiotics, 2020, doi:10.3390/antibiotics9120908_

Round 1

Reviewer 1 Report

Thank you for submitting this nice work to Antibiotics. This paper aimed at studying the biofilm development on aligner materials. Given the increasing trend for patients to prefer such aesthetic aligners, it is essential to investigate potential variations in biofilm adhesion. This will help inform clinicians on optimal materials and advise patients accordingly on enhanced oral hygiene measures. This paper may also stimulate dental material scientists to develop novel materials that can inhibit biofilm assembly. Since the materials tested in this study all had the same chemistry (PETG), a stringer hypothesis is needed to test many types of aligners - why do you think there could be variations in biofilm adhesion? What motivated this study? 

I wonder why the authors use the term "proband" for the patients chosen for saliva collection. The study design (especially the section 4.3.1 is unclear). What do the authors mean by (line 243) "...removed from the labial surface of the central incisors"? 

As is well known, oral biofilms contain pathogens and commensals. Please discuss in greater detail why no species-level analysis or biofilm metabolic activity was performed. 

Author Response

Reviewer 1

  • Thank you for submitting this nice work to Antibiotics. This paper aimed at studying the biofilm development on aligner materials. Given the increasing trend for patients to prefer such aesthetic aligners, it is essential to investigate potential variations in biofilm adhesion. This will help inform clinicians on optimal materials and advise patients accordingly on enhanced oral hygiene measures. This paper may also stimulate dental material scientists to develop novel materials that can inhibit biofilm assembly. Since the materials tested in this study all had the same chemistry (PETG), a stringer hypothesis is needed to test many types of aligners - why do you think there could be variations in biofilm adhesion? What motivated this study? 

Response:

We want to thank the reviewer for this question. Since we tested PETG-based surfaces with the same chemistry, but different surface characteristics, the following sentences were added to the Introduction and Discussion:

Introduction: “Since it Thus, this study aims to examine the initial microbial attachment and biofilm formation on various thermoplastic aligner materials with the same chemistry, but different surface characteristics.”

Discussion: “Since is well known that smooth surfaces with an surface roughness (Sa) threshold of 0.2 μm inhibit bacterial adhesion, we decided to test PETG-based surfaces with similar Ra values (8.1-9.6 nm) in order to investigate if differences in other surface characteristics e.g. Sz could affect microbial adhesion and biofilm formation on the smooth PETG-based surfaces. This assumption was not confirmed in our study.”

  • I wonder why the authors use the term "proband" for the patients chosen for saliva collection.

Response:

We want to make clear that only healthy volunteers and not patients participated in the present study. The term “proband” has been now replaced by the term “participant”. The text describing the selection of study participants has been modified as follows:

“Paraffin-stimulated saliva samples of six healthy volunteers (25 ml from each participant) of both sexes (30-35 years) were collected and pooled. Participants fulfilling one or more of the following criteria were not included in the study: smoking, infectious or severe systemic e.g. salivary gland diseases, presence of carious lesions or periodontal disease, pregnancy or breastfeeding”.

  • The study design (especially the section 4.3.1 is unclear). What do the authors mean by (line 243) "...removed from the labial surface of the central incisors"?

Response:

To enhance clarity the term “labial” has been replaced by the term “frontal” as follows:

“Enamel cylinders (diameter, 4.5 mm; 27.2 mm2 surface area; height, 0.8 mm) were prepared out of the frontal surfaces of the bovine incisors of freshly slaughtered 2-year-old cattle using a water cooled diamond core drill.”

  • As is well known, oral biofilms contain pathogens and commensals. Please discuss in greater detail why no species-level analysis or biofilm metabolic activity was performed. 

Response:

Thank you for this comment the following sentences were added to the manuscript:

“Since the main goal of the study was to examine the initial microbial adhesion and biofilm formation on different PETG-based aligner materials, the determination of CFUs was the best standard method to quantify the total cell count of the viable adherent microorganisms without focusing on specific bacterial species.”

“Further investigation of the biofilm characteristics on the PETG-based surfaces involve culture-based and molecular analyses of the microbial communities on species level as well as the biofilm metabolic activity.”

Reviewer 2 Report

The reviewer's main concern was why each group had 6 samples only. Please justify this by reporting the sample size calculation. The group differences might become significant if the sample size was increased. This also relates to my comment to Figure 1 below.

Figure 1: It is odd to see lower quartile of ERK approached zero. It is not reasonable to observe zero CFU count after 2h. Was particular samples defective? If yes, sample size should be increased or replaced. The authors need to justify for this data.

Stats: Two-way ANOVA was used to test for the group x time effects. It has the assumption of data normal distribution and equal (or at least similar) variances between groups. Figure 1 seemed to suggest the other way round. Please report more data regarding normality test and variance.

Conclusion: Caries and periodontitis were mentioned. If so, the authors should present data particularly concerning the species causing these diseases, e.g. P. Gingivalis. Otherwise please mention this in the limitations or future perspectives that particular strains would be further examined in the future.

Figure: Please add a figure to show readers the specimens.

Author Response

Reviewer 2

  • The reviewer's main concern was why each group had 6 samples only. Please justify this by reporting the sample size calculation. The group differences might become significant if the sample size was increased. This also relates to my comment to Figure 1 below.

Response:

Thank you for the question. Similarly to relevant publications, we tested a total of 96 samples, 16 samples for each group for the two different time points (2h, 72h). The following sentence was added to the manuscript to enhance clarity:

“A total of 96 samples were tested in two different runs (8 samples for each material group in each time point)”.

  • Figure 1: It is odd to see lower quartile of ERK approached zero. It is not reasonable to observe zero CFU count after 2h. Was particular samples defective? If yes, sample size should be increased or replaced. The authors need to justify for this data.

Response:

Thank you for the question. We want to make clear that the samples were not defective. As seen in similar own publications (e.g. Vollmer et al, Antimicrobial photoinactivation using visible light plus water-filtered infrared-A (VIS + wIRA) and Hypericum perforatum modifies in situ oral biofilms, Sci Rep. 2019 Dec 30;9(1):20325) it is possible to have zero values describing the initial adhesion. The zero values can be attributed to the presence of mainly non- or loosely adherent microorganisms on the tested PETG-based surfaces, which are extremely smooth with low Sa values (8.1-9.6 nm) inhibiting initial microbial adhesion.

The following sentence was added to the text:

“The low (even zero) CFU values during initial adhesion can be attributed to the presence of mainly non- or loosely adherent microorganisms on the tested PETG-based surfaces, which are extremely smooth with low Sa values (8.1-9.6 nm) inhibiting initial microbial adhesion.”

  • Stats: Two-way ANOVA was used to test for the group x time effects. It has the assumption of data normal distribution and equal (or at least similar) variances between groups. Figure 1 seemed to suggest the other way round. Please report more data regarding normality test and variance.

Response:

Thank you for the question. Normality test was performed and the following sentences were added under Statistical analysis:

“The Kolmogorov-Smirnov test was applied to investigate the normality of data distribution. Since the data were not normally distributed, Kruskal-Wallis test was used to evaluate the differences between the controls and the experimental material groups.”

  • Conclusion: Caries and periodontitis were mentioned. If so, the authors should present data particularly concerning the species causing these diseases, e.g. P. Gingivalis. Otherwise please mention this in the limitations or future perspectives that particular strains would be further examined in the future.

Response:

Thank you for this comment the following sentences were added to the manuscript (see also response to Reviewers 1,2):

“Since the main goal of the study was to examine the initial microbial adhesion and biofilm formation on different PETG-based aligner materials, the determination of CFUs was the best standard method to quantify the total cell count of the viable adherent microorganisms without focusing on specific bacterial species.”

“Further investigation of the biofilm characteristics on the PETG-based surfaces involve culture-based and molecular analyses of the microbial communities on species level as well as the biofilm metabolic activity.”

  • Figure: Please add a figure to show readers the specimens.

Response:

Although it would be nice to provide a picture with the specimens, we unfortunately lack photos of them. Instead, their dimensions have been described in detail in Materials and Methods as follows:

“Round transparent specimens (diameter, 3 mm; width, 1 mm) were gained from the received disks with a metallic puncher made of a hardened steel.”

Reviewer 3 Report

The manuscript entitled ‘Initial bacterial adhesion and biofilm formation on aligner materials’ by Tektas et al., reports colony counts of aerobic or anaerobic bacteria dislodged from the surfaces of commercially available oral-PETG samples from four different manufactures after in vitro-exposure to normal human saliva. The study confirms safety to use the commercially available aligners. Although the manuscript reviews well on the general knowledge of biofilms in the oral cavity, the results of the study do not give a lot of informative data to understand the mechanisms underlying construction of biofilm on aligner materials or the clinical contribution of biofilms on aligners.

Major concerns:

1. The authors have explained in the introduction Line 64-68, that the surface structure including morphology, chemistry, and charge are mainly responsible for biofilm formation. The study measures biofilm formation on PETG, together with enamel and brackets as comparable controls, but lacks negative or positive control materials for the measurement.

2. The study is limited since it measures only colony counts to present the total bacterial load attached on the material surfaces. Since bacteria race for the surface within seconds after exposure to body fluids, naturally the number of attached bacteria corresponds to the size of the surface area. There could be a difference in the type of bacteria (genus/species/strains) favorable in binding to different surfaces, but that is not investigated in the present study.

3. Whether selection of material affects pathogenic bacteria recruitment in the oral cavity is not determined by the study, since the saliva used are derived from healthy volunteers only. Therefore, it is not certain whether the conclusion ‘suitable for clinical use’ in Line 297 can be justified by the binding capacity of total bacterial counts.

Minor concerns:

1. Table 1 should be included in the Materials and Methods section.

2. Material and Methods section should be placed before the Results section.

3. Methods for determining the surface parameters or any references for the data in Table 1 are missing.

Author Response

Reviewer 3

The manuscript entitled ‘Initial bacterial adhesion and biofilm formation on aligner materials’ by Tektas et al., reports colony counts of aerobic or anaerobic bacteria dislodged from the surfaces of commercially available oral-PETG samples from four different manufactures after in vitro-exposure to normal human saliva. The study confirms safety to use the commercially available aligners. Although the manuscript reviews well on the general knowledge of biofilms in the oral cavity, the results of the study do not give a lot of informative data to understand the mechanisms underlying construction of biofilm on aligner materials or the clinical contribution of biofilms on aligners.

Major concerns:

  1. The authors have explained in the introduction Line 64-68, that the surface structure including morphology, chemistry, and charge are mainly responsible for biofilm formation. The study measures biofilm formation on PETG, together with enamel and brackets as comparable controls, but lacks negative or positive control materials for the measurement.

Response:

Thank you for your question. As in similar studies, enamel serves as a control for the comparison of the natural tooth surface with the PETG-based surfaces. Brackets represent the conventional orthodontic surfaces and are therefore suitable controls for comparison with modern orthodontic materials, namely the PETG-based surfaces. The use of negative or positive controls was not performed, since they are not relevant for clinical use in the oral cavity.

  1. The study is limited since it measures only colony counts to present the total bacterial load attached on the material surfaces. Since bacteria race for the surface within seconds after exposure to body fluids, naturally the number of attached bacteria corresponds to the size of the surface area. There could be a difference in the type of bacteria (genus/species/strains) favorable in binding to different surfaces, but that is not investigated in the present study.

Response:

Thank you for this comment the following sentences were added to the manuscript (see also response to Reviewers 1,2):

“Since the main goal of the study was to examine the initial microbial adhesion and biofilm formation on different PETG-based aligner materials, the determination of CFUs was the best standard method to quantify the total cell count of the viable adherent microorganisms without focusing on specific bacterial species.”

“Further investigation of the biofilm characteristics on the PETG-based surfaces involve culture-based and molecular analyses of the microbial communities on species level as well as the biofilm metabolic activity.”

The size of the surface area was considered after the determination of the CFUs and the graphs 1 and 2 show the CFUs on a logarithmic scale (log10) per cm2.

  1. Whether selection of material affects pathogenic bacteria recruitment in the oral cavity is not determined by the study, since the saliva used are derived from healthy volunteers only. Therefore, it is not certain whether the conclusion ‘suitable for clinical use’ in Line 297 can be justified by the binding capacity of total bacterial counts.

 Response:

Since aligner materials are designed to be used by orthodontic patients, who have to be caries-free and without gingival inflammation prior to and during orthodontic therapy, the choice of healthy volunteers is a realistic simulation of the clinical situation.

Minor concerns:

  1. Table 1 should be included in the Materials and Methods section.

Response:

Table 1 has been renumbered as Table 2 and is included in Materials and Methods. The following sentence was added to the text:

“Table 2 shows the surface different surface parameters of the tested aligner materials., which were described in a recent own study [23].”

  1. Material and Methods section should be placed before the Results section.

Response:

According to the “Instructions for authors” of the journal the section of “Materials and Methods” should be written after the Discussion. Additionally, we used a word-template provided by the journal.

  1. Methods for determining the surface parameters or any references for the data in Table 1 are missing.

Response:

The determination of the surface parameters in included in the following report:

Suter, F.; Zinelis, S.; Patcas, R.; Schätzle, M.; Eliades, G.; Eliades, T. Roughness and wettability of aligner materials. J Orthod 2020, 47, 223-231, doi:10.1177/1465312520936702.

The following sentence was added to the text (see also response to minor concern 1):

“Table 2 shows the surface different surface  parameters of the tested aligner materials., which were described in a recent own study [23].”

Reviewer 4 Report

Bacterial Adhesion and Biofilm Formation on Aligner Materials

1. "This protocol was applied three times, obtaining an average of nine independent measurements for each material group." This suggests that each material was used in triplicate for each of the three runs (3 x 3 = 9). "Six samples of each material group were incubated in saliva for 2 and 72 hours." This should be Three samples of each material group were incubated in saliva for 2 and 72 hours.

2. "A total of 96 samples were tested in two different runs (8 samples for each material group in each time point)." This is confusing. How do you get 96 samples? Were the enamal and bracket controls not included in the three runs with the other materials? Please clarify.

3. line 79: "Figure 1A shows the results of the initially adherent oral aerobic and anaerobic microorganisms ..."

Figure 1A shows results for aerobic microorganisms. Figure 1B shows the results for the anaerobes.

4. line 165: " ... with an surface ..." --> with a surface

5. line 230: " ... collected by 6 donors ..." --> collected from 6 donors

6. line 232: six --> three

7. line 239: systemic eg. --> systemic condition, eg.

8. line 243: Convert rpm to x g.

9. section 4.2: Was informed consent obtained from the 6 subjects?

10. line 287: "thawed saliva" --> thawed microbial suspension

11. line 290: "saliva suspension" --> microbial suspension

12. line 295: "saliva suspensions" --> microbial suspensions

13. Table 2: What do Sa, Sz Sq, Sdr and Sv represent.  Add this to the legend.

Author Response

Reviewer 4

  1. "This protocol was applied three times, obtaining an average of nine independent measurements for each material group." This suggests that each material was used in triplicate for each of the three runs (3 x 3 = 9). "Six samples of each material group were incubated in saliva for 2 and 72 hours." This should be Three samples of each material group were incubated in saliva for 2 and 72 hours.

Response:

Thank you for the question. The sentences were adjusted as follows:

“Two individual experiments were performed, and each group was represented in quadruplicate in each experiment.”

Four samples of each material group were incubated in saliva for 2 and 72 hours, respectively.

  1. "A total of 96 samples were tested in two different runs (8 samples for each material group in each time point)." This is confusing. How do you get 96 samples? Were the enamal and bracket controls not included in the three runs with the other materials? Please clarify.

Response:

Thank you again for this question. Please see the previous response. The text has been changed accordingly.

  1. line 79: "Figure 1A shows the results of the initially adherent oral aerobic and anaerobic microorganisms ..."

Figure 1A shows results for aerobic microorganisms. Figure 1B shows the results for the anaerobes.

Response: Thank you. The text was corrected accordingly.

  1. line 165: " ... with an surface ..." --> with a surface

Response: Thank you. The text was corrected accordingly.

  1. line 230: " ... collected by 6 donors ..." --> collected from 6 donors

Response: Thank you. The text was corrected accordingly.

  1. line 232: six --> three

Response: Thank you. The text was corrected accordingly. See our previous response to your first question.

  1. line 239: systemic eg. --> systemic condition, eg.

Response: Thank you. The text was corrected accordingly.

  1. line 243: Convert rpm to x g.

Response: Thank you, but we prefer to use revolutions per minute (rpm).

  1. section 4.2: Was informed consent obtained from the 6 subjects?

Response: Since the pooled saliva was used, an informed consent was not necessary, according to the guidelines of local Ethics Committee.

  1. line 287: "thawed saliva" --> thawed microbial suspension

Response: Thank you. The text was corrected accordingly.

  1. line 290: "saliva suspension" --> microbial suspension

Response: Thank you. The text was corrected accordingly.

  1. line 295: "saliva suspensions" --> microbial suspensions

Response: Thank you. The text was corrected accordingly.

  1. Table 2: What do Sa, Sz Sq, Sdr and Sv represent.  Add this to the legend.

Response: Thank you. The table legend was changed as follows: “Table 2. Mean values and standard deviations (in parentheses) of five surface roughness parameters, namely average surface roughness (Sa), ten-point average roughness (Sz), root mean square surface roughness (Sq), developed surface area ratio (Sdr), and maximum pit height (Sv). The superscript letter a denotes values without statistical significant differences per parameter within and between different treatment groups (p>0.05).”

Round 2

Reviewer 3 Report

Major concern 1.

The authors were not able to give an explanation as to why experimental measurement controls in the colony count-analysis on polymer surfaces were not included.

Major concern 2 and 3.

The conclusion of the current study simply justifies usage of aligners that are already commercially available.

If there is to be a study reflecting bacterial colony formation on the devices, determining the composition of bacteria within the biofilm or to explore on whether any of the polymers chosen could serve as a biomaterial surface in the oral cavity for bacteria with a pathogenic character, or to see whether the bacteria may have an advanced effect on the polymers, such as for example, deterioration of the device, could be of clinical importance. Unfortunately, the study does not give scientific information required for biofilm research.

Author Response

Reviewer 3

  • Major concern 1.

The authors were not able to give an explanation as to why experimental measurement controls in the colony count-analysis on polymer surfaces were not included.

 Response:

As commented in our previous response, controls with clinical/biological significance were indeed used in this study. Specifically, enamel served as a control for the comparison of the natural tooth surface with the PETG-based surfaces. Brackets represented the conventional orthodontic surfaces and are therefore suitable controls for comparison with modern orthodontic materials, namely the PETG-based surfaces. The surface characteristics of the tested aligner surfaces were reported in a recent publication (Suter, F.; Zinelis, S.; Patcas, R.; Schätzle, M.; Eliades, G.; Eliades, T. Roughness and wettability of aligner materials. J Orthod 2020, 47, 223-231, doi:10.1177/1465312520936702).

To confirm our assumptions, we would like the following text was added to the manuscript: “Enamel served as a control for the comparison of the natural tooth surface with the PETG-based surfaces. Brackets represented the conventional orthodontic surfaces and are therefore suitable controls for comparison with modern orthodontic materials, namely the PETG-based surfaces. ”

  • Major concern 2 and 3.

The conclusion of the current study simply justifies usage of aligners that are already commercially available.

Response: To our knowledge, this is the first study to assess the initial bacterial adhesion and biofilm formation on the four different aligner materials. Although the aligner materials are commercially available, no study reporting the favoritism of initial microbial adhesion and biofilm formation on the PETG-surfaces was found. However, the surface characteristic of the aligners have been tested in a previous study as mentioned above (Suter, F.; Zinelis, S.; Patcas, R.; Schätzle, M.; Eliades, G.; Eliades, T. Roughness and wettability of aligner materials. J Orthod 2020, 47, 223-231, doi:10.1177/1465312520936702). Since we tested PETG-based surfaces with the same chemistry, but different surface characteristics, this study aims to examine the initial microbial attachment and biofilm formation on various thermoplastic aligner materials with the same chemistry, but different surface characteristics. Since is well known that smooth surfaces with an surface roughness (Sa) threshold of 0.2 μm inhibit bacterial adhesion, we decided to test PETG-based surfaces with similar Ra values (8.1-9.6 nm) in order to investigate if differences in other surface characteristics e.g. Sz could affect microbial adhesion and biofilm formation on the smooth PETG-based surfaces. This assumption was not confirmed in our study.

The following sentence was added to the discussion: “To our knowledge, this is the first study to assess the initial bacterial adhesion and biofilm formation on the four different aligner materials, which are commercially available.”

  • If there is to be a study reflecting bacterial colony formation on the devices, determining the composition of bacteria within the biofilm or to explore on whether any of the polymers chosen could serve as a biomaterial surface in the oral cavity for bacteria with a pathogenic character, or to see whether the bacteria may have an advanced effect on the polymers, such as for example, deterioration of the device, could be of clinical importance. Unfortunately, the study does not give scientific information required for biofilm research.

Response:

Thank you for the question. The following text was added to the manuscript:

“The changes accompanying the intraoral service of aligners have been reported for a wide array of materials including PETG and polyurethane-based polymers, with respect to mechanical properties and surface roughness in recent research from our group. (Changes in Roughness and Mechanical Properties of Invisalign Appliances after One- and Two-Weeks Use. Papadopoulou AK, Cantele A, Polychronis G, Zinelis S, Eliades T.  Materials (Basel). 2019 Jul 28;12(15):2406. doi: 10.3390/ma12152406.)

Because of the relatively short exposure time of aligners to the intraoral environment, the imposed changes are limited to the effects of water on the polymeric material. Hydrolytic degradation and attack of the polyurethanes by water has been described in the biomedical literature as the main mechanism of disintegration of polymers exposed to aqueous media. 

(Polymer Degradation and Stability Volume 97, Issue 8, August 2012, Pages 1553-1561 Hydrolytic degradation of segmented polyurethane copolymers for biomedical applications SubrataMonda DarrenMartin).”

The statement of the reviewer that bacterial may play a decisive role in the disintegration of aligners lacks a substantiation and cannot serve as a working hypothesis to postulate potential effects of the colonies formed on surface on material surface alterations. It is worth noting however, that the objective of the study was not to assign to microbial a disintegrating role on polymers as microbial concentration on aligners may have an effect on the hard tissue (enamel in that case) decalcification arising from the presence of Streptococcus mutans by-products.